# A Concise Review on Electrospun Scaffolds for Kidney Tissue Engineering

**DOI:** 10.3390/bioengineering9100554

**Published:** 2022-10-14

**Authors:** Cláudia C. Miranda, Mariana Ramalho Gomes, Mariana Moço, Joaquim M. S. Cabral, Frederico Castelo Ferreira, Paola Sanjuan-Alberte

**Affiliations:** 1Department of Bioengineering, Institute for Bioengineering and Biosciences, Instituto Superior Técnico, Universidade de Lisboa, Av. Rovisco Pais, 1049-001 Lisbon, Portugal; 2Associate Laboratory i4HB—Institute for Health and Bioeconomy, Instituto Superior Técnico, Universidade de Lisboa, Av. Rovisco Pais, 1049-001 Lisbon, Portugal

**Keywords:** kidney organoids, kidney tissue engineering, electrospinning, iPSCs, disease modelling, developmental biology

## Abstract

Chronic kidney disease is one of the deadliest diseases globally and treatment methods are still insufficient, relying mostly on transplantation and dialysis. Engineering of kidney tissues in vitro from induced pluripotent stem cells (iPSCs) could provide a solution to this medical need by restoring the function of damaged kidneys. However, implementation of such approaches is still challenging to achieve due to the complexity of mature kidneys in vivo. Several strategies have been defined to obtain kidney progenitor endothelial and epithelial cells that could form nephrons and proximal tube cells, but these lack tissue maturity and vascularisation to be further implemented. Electrospinning is a technique that has shown promise in the development of physiological microenvironments of several tissues and could be applied in the engineering of kidney tissues. Synthetic polymers such as polycaprolactone, polylactic acid, and poly(vinyl alcohol) have been explored in the manufacturing of fibres that align and promote the proliferation and cell-to-cell interactions of kidney cells. Natural polymers including silk fibroin and decellularised extracellular matrix have also been explored alone and in combination with synthetic polymers promoting the differentiation of podocytes and tubular-specific cells. Despite these attempts, further work is still required to advance the applications of electrospun fibres in kidney tissue engineering and explore this technique in combination with other manufacturing methods such as bioprinting to develop more organised, mature and reproducible kidney organoids.

## 1. Introduction

Chronic kidney disease (CKD) affects approximately one in ten individuals worldwide, with forecasts suggesting that by 2040 CKD will become the fifth highest worldwide cause of years of life lost [1]. Dialysis and transplantation are still the main therapeutic approaches used in CKD, but are limited by a shortage of donor availability and have a great impact on the patients’ quality of life [2]. Recently, the use of human-induced pluripotent stem cells (hiPSCs) has meant a revolution in organ and tissue engineering (TE) and they hold strong potential in the therapy of CKD.

The major goal of kidney TE is replacing, restoring or enhancing the biological function of damaged kidneys. However, achieving this is still extremely challenging due to the high complexity that mature kidneys present [3]. A step towards investigating and understanding the development of mature kidney tissues is the development of kidney organoids, which can also be employed to study the interactions between different cell types and to identify targets for the development of novel therapeutic approaches [4].

Different kidney cell types have been successfully reproduced in organoids as shown in the following section, but their level of maturity and lack of vascularisation within the organoid still remains a challenge. Further bioengineering approaches are therefore required to design cell microenvironments and emulate physiological cues [5]. Advances in scaffolding materials such as kidney decellularised extracellular matrix (dECM) have improved the maturation of glomerular-like structures [6], and bioprinting approaches are promising to develop organoids in the macroscale [7].

Electrospinning is a technique that has been widely investigated in the engineering of multiple tissues such as bone [8], skin [9] or cardiac tissues [10]. However, it still remains largely unexplored in kidney TE. Nevertheless, the use of electrospun materials is still of great interest in this area as this technique is compatible with multiple materials [11], presents tuneable mechanical properties [12] and allows the release of components with interest in kidney TE such as vascular endothelial growth factor (VEGF) [13] to promote the vascularisation of kidney organoids (Figure 1).

In the following sections of this mini-review, we describe for the first time the current advances made on the application of electrospinning techniques in kidney TE and highlight the potential advantages of such approaches over the use of other bioengineering techniques.

## 2. Kidney Cells Differentiation and Development of Kidney Organoids

### 2.1. Embryonic Development of the Kidney

The kidney is a complex organ that plays a key role in human physiology, being responsible for blood filtration. This process ensures homeostasis maintenance, body fluid composition regulation through water and waste removal, as well as control of electrolytes and non-electrolytes concentration [14,15,16]. 

In mammals, kidney develops from the intermediate mesoderm (IM), progressing through three major stages: pronephros, mesonephros and metanephros. The first two stages are transient whereas metanephric kidney persists as a mature and functional kidney in mammals [17,18]. 

Metanephric kidney derives from the reciprocal interactions of two primordial mesodermal tissues, the epithelial ureteric bud (UB) and the metanephric mesenchyme (MM) [19,20], both generated from the IM through fibroblast growth factor (FGF) and Wnt signalling pathway modulation [21] (Figure 2). The UB gives rise to renal collecting system epithelial cells, whereas MM gives rise to nephron epithelial cells as well as endothelial and stromal cell precursors [22,23]. The functional units of the kidneys, the nephrons, arise from MM cells in a process denominated nephrogenesis.

Nephrogenesis process ceases around the 36th week of gestation [24]. The postnatal human kidney lacks a nephron progenitor population, as this population is terminally differentiated prior to birth [20]. Consequently, the adult kidney is unable to generate new nephrons, which highlights the pressing need to improve our understanding of renal pathologies and to develop strategies that allow us regenerate damaged kidney tissues.

### 2.2. In Vitro Production of Human Kidney Cells and Organoids

Kidney organoids derived from hiPSCs might be attractive three-dimensional (3D) models for different purposes, namely, to model kidney embryonic development, kidney disease, high-throughput drug screening and potential implementation in renal regeneration and replacement therapies [25,26,27]. There have been several successful attempts to derive kidney organoids in vitro, starting from simpler approaches that generate one or a few cell types [28,29,30] to 3D cultures that recreate different structures of the kidney [31,32,33,34]. Primitive kidney morphogenesis has been accomplished through application of a defined cocktail of growth factors and small molecules including CHIR99021, FGF-9 and heparin [32]. However, this approach often yields populations that lack mature phenotypes from the adult kidney, resembling cell types present during the first trimester of gestation [35].

Generation of kidney organoids from hiPSCs, similar to the embryonic kidney development, depends on cell-to-cell interactions between four distinct progenitor cell populations, namely, nephron progenitor cells (NPCs), UB progenitor cells, endothelial progenitor cells, and stromal progenitor cells [23]. In addition to the signalling pathways, the timing, level and duration of a particular signal must be carefully considered, as these variables influence cellular responses [36]. For instance, the duration of Wnt signalling activation determines whether the UB or MM cell population becomes the dominant population [37]. Then UB and MM cell populations interact with one another, resulting in further differentiation and self-organisation into nephron structures [26].

The in vitro derivation of kidney structures provides a bottom-up approach that can lead to an easier strategy when replicating in vivo structures following tissue engineering principles. For that, there is the need to generate specific cell types, such as proximal tubule cells [38] or podocytes [39]—in the case of the kidney—instead of organoids containing various types of cells and defined structures.

Therefore, to replicate the kidney glomerular environment, it is crucial to investigate, not only the cell-to-cell interactions, but also the necessary ECM components, scaffolding materials and physicochemical or mechanical cues that efficiently promote hiPSC differentiation, organisation and overall cell survival [40,41]. Kidney cells’ behaviour is strongly dictated by the complex 3D microenvironment wherein they reside. Therefore, to develop kidney organoids with physiological characteristics, it is key to recreate the kidney cell niche in high detail and as shown below, electrospinning is a technique that could allow us to achieve that.

## 3. Use of Electrospinning in the Development of Kidney Tissues 

### 3.1. Overview of the Electrospinning Techniques

Electrospinning is a simple scaffold fabrication technique that enables the production of continuous fibres in the nano- and micro- range from a wide diversity of materials [42]. During electrospinning, a high electric field is applied to a polymer solution in the tip of a nozzle, which consequently creates a Taylor cone due to the surface tension between the polymer solution and nozzle walls. When the applied electrical force overcomes the surface tension, a jet is formed from the Taylor cone into the direction of the collector and, upon solvent’s evaporation, solidified fibres are formed in the collector [43]. Moreover, the fibre’s diameter can be tailored by carefully controlling several parameters such as nozzle size, applied electrical field, polymer solution and atmosphere [44]. 

Electrospun fibres are of great interest for TE applications since they mimic the morphology of the ECM of native tissues due to their topological dimensions, high surface-to-volume ratio and high pore interconnectivity [42]. The high pore size and pore interconnectivity are essential to promote the diffusion of nutrients, cells and oxygen [45]. 

Some variations of the conventional electrospinning technique are also of interest in kidney TE as they can potentially increase the functionality of the fibres produced. 

For instance, scaffolds with a higher degree of porosity can be obtained by cryogenic electrospinning (CES) in which ice crystals are deposited on a cooled surface acting as a template [46]. The increased porosity can contribute towards a higher cellular infiltration and vascularisation and the development of bi-layered tissue structures [47]. To demonstrate this, a study evaluated the effects on 3T3/NIH cell infiltration of conventional and CES fibres and concluded that the larger pore size influences cell infiltration and attachment [48].

Additional non-conventional electrospinning techniques also allow the manufacturing of multifunctional systems. Coaxial electrospinning is often used to manufacture double-layer fibres with various functionalities, enhanced mechanical properties and incorporate drugs and molecules [49]. Emulsion electrospinning also enables the production of coaxial fibres in which growth factors and proteins can be encapsulated [50]. VEGF and nerve growth factor (NGF) were both delivered to induced pluripotent stem cell-derived neural crest+ stem cells from emulsion electrospun nanofibrous scaffolds contributing towards their neovascularisation and nerve healing [51].

The electrospinning technique is also compatible with live cells, encapsulating them in the desired environment [52]. This could provide a better understanding of the cell-matrix interactions required to build up the spatial organisation of renal cell populations. Some examples of cell electrospinning include the electrospraying of human adipose-derived stem cells in gelatin/pullulan polymer solutions to enhance their differentiation into chondrocytes [53].

Further information on electrospinning techniques and their use in tissue engineering can be found elsewhere [54,55,56].

### 3.2. Electrospun Fibres Used in Kidney Tissue Engineering

Electrospun scaffolds used in kidney TE have been fabricated from synthetic and natural polymers or a combination of both [57]. Synthetic polymers offer higher reproducibility and control regarding mechanical properties. Importantly, they can be tailored to meet the requirements of certain applications [58]. Natural polymers, on the other hand, possess the suitable biochemical cues for cell attachment and proliferation and their degradation by-products are non-toxic. However, the main drawbacks are their poor mechanical properties, crosslinking treatment requirements, lack of control in composition and molecular weight and variability from batch to batch [59], therefore, they are often combined with synthetic polymers.

#### 3.2.1. Synthetic Polymers

The most commonly used synthetic-based electrospun scaffolds in kidney electrospinning are composed of polycaprolactone (PCL), polylactic acid (PLA) and poly(vinyl alcohol) (PVA). 

PCL is one of the most commonly used polymers for biomedical applications. This polymer is biocompatible and biodegrades via hydrolysis of the ester groups [60]. Burton et al. produced novel PCL scaffolds with different architectures and porosities by conventional electrospinning and CES aiming to investigate their influence on the viability and morphology of human kidney epithelial cells. Although PCL is a hydrophobic material, plasma treatment enhanced the hydrophilicity of the scaffolds without compromising their mechanical properties. Moreover, thicker fibres promoted a higher cellular adhesion, viability, and alignment; and an upregulation of the ANPEP gene, a key marker for proximal tubular epithelial cells, was observed [46]. In this case, scaffolds were sterilised in 70% ethanol for an hour followed by washings in distilled water and drying under vacuum for 24 h prior to plasma treatment.

The high stiffness of PCL can be overcome by introducing other polymers within its matrix such as laminin and collagen. Baskapan et al. produced novel scaffolds composed of PCL and laminin by blend and emulsion electrospinning. Interestingly, the laminin turned the scaffold more elastic, promoting more cell-to-cell and cell-to-fibre interactions. Moreover, the expression of the genes E-CAD, KIM-1 and ANPEP was investigated, showing that hybrid scaffolds provide a more suitable environment for kidney cells [12]. E-CAD is a key marker for the formation of cell junctions in epithelial tissues, increasing overtime in all the conditions studied, suggesting the formation of monolayers in the electrospun fibres. KIM-1 is a marker related to acute kidney injury and its expression was lower in scaffolds combining PCL and laminin than in PCL alone [12]. Prior to cell seeding, scaffolds were sterilised in 70% ethanol for 30 min and washed in phosphate buffer saline (PBS).

PLA has been used in the development of drug release systems in cardiac and kidney TE mainly due to its high corrosion resistance, chemical stability, and elastomeric properties [61]. Burton et al. showed that PLA fibres sterilised in 70% isopropanol were biocompatible with cells from the proximal tubules, collecting duct, glomerular epithelia and glomerular endothelia, regardless the fibre diameter [62]. Nevertheless, PLA is hydrophobic and brittle, which hinders cellular adhesion and growth and its implementation in applications with high plastic deformation [63]. To overcome this, Alharbi et al. used coaxial electrospinning to produce scaffolds with PLA in the core and PVA in the shell subsequently heated in a furnace at 50 °C [63]. No further pre-treatment or sterilisation methods are mentioned in this work, where coaxial electrospun fibres supported the attachment of human embryonic kidney cells (HEK-293).

PVA-based scaffolds have gained interest in TE applications due to their high biocompatibility, thermal stability and chemical resistance. Moreover, the polymer’s molecular weight influences its thermal stability and mechanical properties. Despite the advantages, the use of PVA requires crosslinking treatments that can be accomplished by chemical or physical reactions. For instance, physical methods including ultraviolet radiation and freezing/thawing are non-toxic and therefore, more suitable for TE applications [64]. The previous study showed that scaffolds promoted the proliferation and adhesion of human embryonic kidney cells. For instance, PVA has outstanding surface wetting and mechanical properties, being a good candidate for kidney TE [63]. PVA-based electrospun fibres with a controlled release of silver nanoparticles have also shown promise to support human embryonic kidney cells (293T) while maintaining a strong antimicrobial activity against Gram-positive *S. aureus* [65]. 

#### 3.2.2. Natural Polymers

Silk fibroin (SF) is a natural macromolecular protein polymer promising for TE applications due to its high biocompatibility, biodegradability and outstanding mechanical properties. For instance, the hydrophobic β-sheet-forming domains present on SF contribute to an increased resilient behaviour [66]. For the first time, Mou et al. successfully produced SF fibres through electrospinning that promote the differentiation of mature human podocytes. Furthermore, the manufactured scaffold was able to support long-term culture and cellular viability as well as an upregulated expression of the podocyte-specific markers PAX2, NPHS1 and WT1 and comparable levels of PODXL, SYNPO and NEPH1 to cells cultured on laminin-coated plates [66]. Prior to cell culture, samples were plasma treated, sterilised in 70% ethanol, washed in sterile water and coated with 25 µg mL^−1^ laminin-511.

Another valid strategy to develop scaffolds for cells relies on decellularisation. Decellularised ECM (dECM) is enriched in proteins and glycoproteins, providing a native kidney-like environment. Therefore, scaffolds composed of dECM can trigger several cellular activities (e.g., attachment, differentiation, maturation) [60]. Recently, Sobreiro-Almeida et al. produced novel scaffolds composed of dECM and PCL through electrospinning for reproducing the tubular basement membrane. The biological assays were carried out in vitro where the scaffolds promoted a greater metabolic activity, proliferation and protein content of human renal progenitor cells (hRPCs). Moreover, the tubular-specific markers SCL12A1, SLC3A1, SLC9A3 and GGT1 were detected. The scaffold was also validated as an epithelium-endothelium model on co-cultures containing differentiated epithelial tubular cells (hRPTECs) and human umbilical vein endothelial cells (HUVECs), demonstrating that the uptake of human serum albumin is higher in co-cultures than in monocultures [67]. Sterilisation of these samples consisted of immersion in 70% ethanol and 30 min exposure to ultra-violet (UV) radiation.

## 4. Future Prospects

Despite being a technique commonly used in the engineering of multiple tissues, electrospinning has still not been widely explored in kidney TE, an area where cell bioprinting has been the focus [60]. On the other hand, several renal reabsorption models have been proposed in the literature, however these are often associated with the use of immortalised cell lines that may fail to fully recapitulate physiological features [68]. Therefore, as electrospun fibres have been shown to recapitulate ECM features, models combining renal derivation and electrospinning may provide a suitable alternative for generating organ-specific structures to employ in functional studies and drug screening/discovery assays. Furthermore, manufacturing of scaffolds by electrospinning offers flexibility on materials used and scaffold architecture, potential biofunctionalisation of electrospun fibres and possibility to incorporate biomolecules, drugs and other compounds of interest for sustained release, particularly when non-conventional electrospinning techniques are used. This could be employed to control differentiation and maturation of kidney cells, such as proximal tubule cells, more accurately over time [38]. In order to further advance this area, it would be interesting to explore additional materials and composites to the ones described herein.

Development of precise and complex kidney organoids would also potentially represent the first step towards applying this technology in the regeneration of damaged areas of the kidney and electrospinning could be key to achieving this. In previous studies, the surface topography of electrospun fibres controlled the cell migration and alignment of stem cells to form specific organoids [69]. Therefore, the combination of various fabrication methods such as bioprinting and electrospinning could be useful to generate organoids with defined architectures and size, improving their reproducibility and regulation.

## 5. Conclusions

Despite the progress experienced over the past decades, achieving mature kidney tissues in vitro is still challenging, in part due to the lack of knowledge in the specific stimuli required for the development of kidney organoids. Further development on this field is of particular interest for its potential in a plethora of applications ranging from regenerative medicine to development of kidney-on-chips for drug screening, disease modelling and developmental biology.

In this short review, application and current trends of electrospinning in kidney TE is presented, which is still quite limited despite being applied in the engineering of other tissues such as bone, neural and cardiac tissues. Several synthetic and natural polymers—and a combination of these—have been explored, leading to more suitable microenvironments compatible with multiple kidney cell types and kidney stem cell precursors.

Further work in this area could offer significant advances in kidney TE that lead to clinical translation, therefore, there are multiple opportunities in this field awaiting to be explored.

## Figures and Tables

**Figure 1 bioengineering-09-00554-f001:**
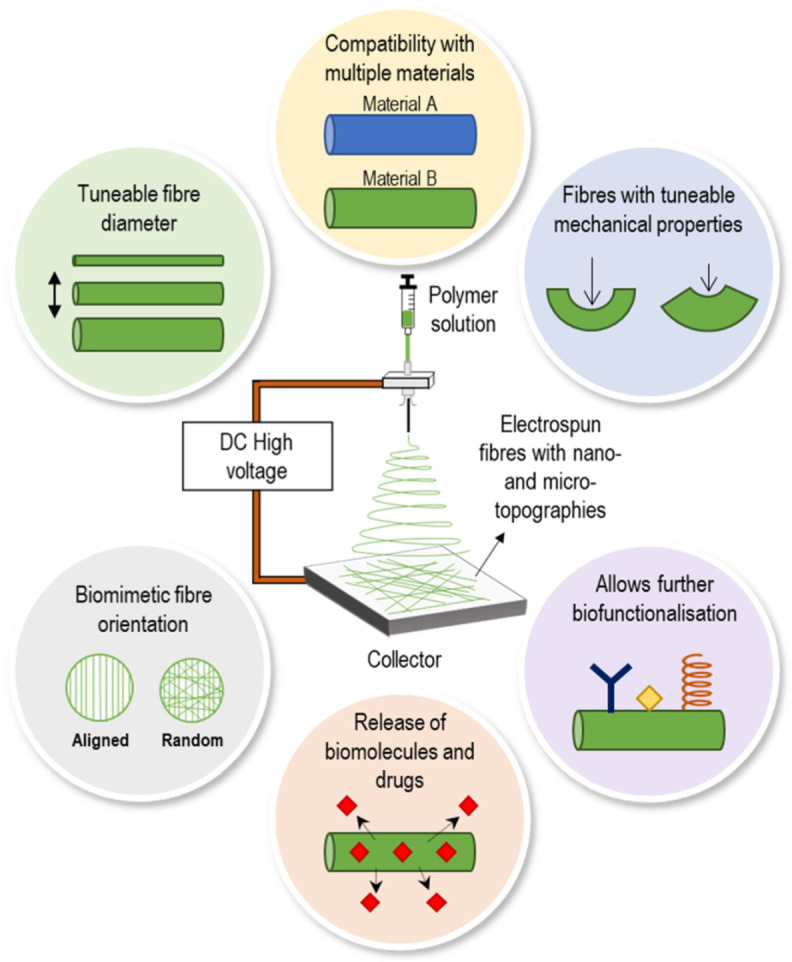
Schematic of an electrospinning setup and description of some of its features with interest in kidney tissue engineering.

**Figure 2 bioengineering-09-00554-f002:**
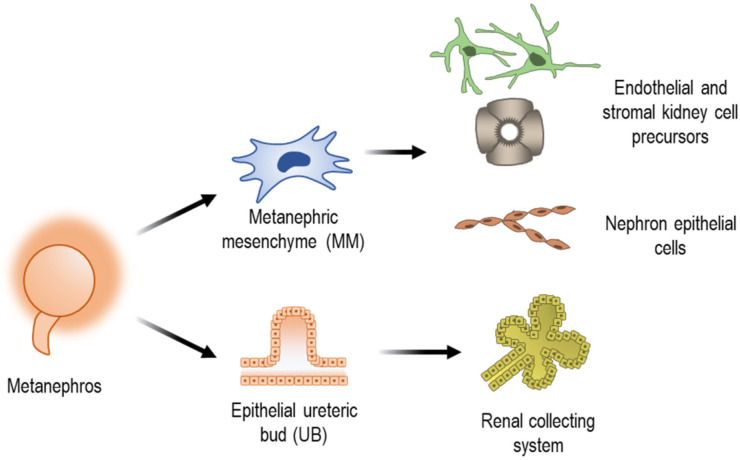
Schematic showing the derivation of metanephros into different kidney cell types.

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
