# Peer review of "A Concise Review on Electrospun Scaffolds for Kidney Tissue Engineering"

_bioengineering, 2022, doi:10.3390/bioengineering9100554_

Round 1
Reviewer 1 Report
Thank you for your paper, interesting but most of the text is repetitive and it’s been done before.
I suggest adding a section to make it different from other review papers.
Please add some more to line 52 since there are new lines of electrospinning like live cell electrospinning and cite some papers like below:
Nosoudi, Nasim, et al. "Differentiation of adipose-derived stem cells to chondrocytes using electrospraying." Scientific reports 11.1 (2021): 1-11.
Jayasinghe, Suwan N. "Cell electrospinning: a novel tool for functionalising fibres, scaffolds and membranes with living cells and other advanced materials for regenerative biology and medicine." Analyst 138.8 (2013): 2215-2223.
Nosoudi, Nasim, et al. "Electrospinning live cells using gelatin and pullulan." Bioengineering 7.1 (2020): 21.
You need to talk about electrospraying and spinning in section” 3.1. Description of the electrospinning technique” extensively as a potential future approach to electrospin kidney cells.
Author Response
Dear reviewer,
We appreciate the comments and changes have been done to the text to avoid repetition and expand some sections as suggested. To answer to your specific comments, we have done the following changes to the text:
- More detailed has been provided on non-conventional electrospinning techniques in section 3.1.
- Cell electrospinning and electrospraying have also been discussed, including the suggested references
Reviewer 2 Report
The authors have provided a mini-review examining the potential of electrospun scaffolds as a means of advancing culture conditions for tissues originating from physiological systems. Specifically, the authors highlight the limitations of certain culture conditions for tissues/cell sensitive to their surrounding environment, and identify the potential of electrospun scaffolds to create physical environments with the capacity to be functionalised for advancing the models we currently rely on for engineering and medicinal studies. Overall, this is an interesting and pertinent topic.
In reviewing the article however I had some concerns. The following hould be addressed by the authors when preparing a suitable revision.
1. Under section 3.2, it would be advised to use numbered subheadings for ‘synthetic polymers’ and ‘natural polymers’ to distinguish them.
2. Overall, the writing is quite strong but there are instances of typos/grammatical errors within. The authors should review the manuscript and eliminate them accordingly.
3. Perhaps the greatest limitation of this review is the level of detail in which the authors explore certain aspects of the topic. Granted, this is a mini-review, but at times much wordage is wasteful and repetitious at times, while some points are underdeveloped and raise more questions than act as informative. For example, in ‘Synthetic Polymers’ the authors state that ‘key genes was investigated’ – what ‘key genes’? Name them, give some context. This is throughout the review, and the authors should examine this and reconfigure some writing to be more informative in its nature.
4. It would be interesting to know how these materials were treated for sterility prior to use in cell culture. Some drawbacks/problems of these materials are given from the material perspective, but it would be more balanced if this was given from a cell culture application perspective.
Author Response
Dear reviewer,
your comments are much appreciated and we have introduced the following changes to the text to address your comments:
- Under section 3.2, it would be advised to use numbered subheadings for ‘synthetic polymers’ and ‘natural polymers’ to distinguish them.
Subheadings have been added on both sections
- Overall, the writing is quite strong but there are instances of typos/grammatical errors within. The authors should review the manuscript and eliminate them accordingly.
Manuscript has been revised to correct any existing typos
- Perhaps the greatest limitation of this review is the level of detail in which the authors explore certain aspects of the topic. Granted, this is a mini-review, but at times much wordage is wasteful and repetitious at times, while some points are underdeveloped and raise more questions than act as informative. For example, in ‘Synthetic Polymers’ the authors state that ‘key genes was investigated’ – what ‘key genes’? Name them, give some context. This is throughout the review, and the authors should examine this and reconfigure some writing to be more informative in its nature.
More specific details and context have been provided to the different sections, particularly on sections 3.1 and 3.2, where non-conventional electrospinning techniques and their potential in kidney TE have been discussed more deeply and information on the biological outputs has been expanded.
- It would be interesting to know how these materials were treated for sterility prior to use in cell culture. Some drawbacks/problems of these materials are given from the material perspective, but it would be more balanced if this was given from a cell culture application perspective.
Details on how the different materials were pre-treated before cell culture have now been included in the text.
Reviewer 3 Report
The work entitled “A concise review on electrospun scaffolds for kidney tissue engineering” reports on the current advances triggered by the use of electropunk scaffolding in kidney tissue engineering and the potential advantages introduced by this technique. As the authors refer in the title, the work is very concise, direct and mini. Even though the subject is up to date, the information provided is so reduced that we take an actual conclusion. Is this all the research the authors could find on the subject. It is precise, for sure, but as a rule a review offers a bit more detail on the subject which is not given. I would recommend the authors to further improve this work, offering even an evolution on the technologies that have been used of late and how electrospinning improves over those. Regardless, the work has merit and is very well written and should be considered for publication. In fact, I recommend its publication after major revision are implement, by the addition of this new section.
Author Response
Dear reviewer,
We appreciate the feedback provided. To answer to your comments, while there is not much literature on this topic, we have rewritten the sections of the mini-review to expand the level of details on them, particularly on sections 3.1 and 3.2, where non-conventional electrospinning techniques and their potential in kidney TE have been discussed more deeply and information on the biological outputs has been expanded.
Round 2
Reviewer 1 Report
Thanks for addressing my concerns.
Reviewer 2 Report
The authors have responded positively to my previous comments, and their actions have seen the manuscript become much improved.
Reviewer 3 Report
The authors have implemented significant alterations in the manuscript and it is now ready for publication.